



# Mapping of peatlands in the forested landscape of Sweden using LiDAR-based terrain indices

Lukas Rimondini[1], Thomas Gumbricht[1], Anders Ahlström[2], Gustaf Hugelius[1]

[1] Department of Physical Geography, Stockholm University, Stockholm, 10691 Stockholm, Sweden
[2] Department of Physical Geography and Ecosystem Science, Lund University, 22362 Lund, Sweden

*Correspondence to*: Lukas Rimondini (lukas.rimondini@natgeo.su.se)

**Abstract.** Globally, northern peatlands are major carbon deposits with important implications for the climate system. It is therefore crucial to understand their spatial occurrence, especially in the context of peatland degradation by land cover and climate change. This study was aimed at mapping peatlands in the forested landscape of Sweden by modelling soil data against LiDAR-based terrain indices. Machine learning methods were used to produce nation-wide raster maps at 10 m spatial resolution indicating presence-or-not of peatlands. Four different definitions of peatlands were examined: 30, 40, 50 and 100 cm thickness of the organic horizon. Depending on peatland definition, testing with a hold-out dataset indicated Accuracy of 0.89 - 0.91 and Matthew's correlation coefficient of 0.79 - 0.81. The final maps showed a national forest peatland extent of 60,726 - 72,604 km2, estimates which are in range with previous studies employing traditional soil maps. In conclusion, these results emphasize the possibilities of mapping boreal peatlands with LiDAR-based terrain indices. The final peatland maps are publicly available and may be employed for spatial planning, estimating carbon stocks and evaluate climate change mitigation strategies.

## 1 Introduction

Peatlands support biodiversity, buffer hydrological cycles and sequester and store atmospheric carbon (C). They have historically been important C sinks and it is estimated that peatlands in the northern hemisphere hold $415 \pm 150$ Pg C (Hugelius et al, 2020). As peatlands are degraded by climate and land cover change, this long-term C sink is however becoming a substantial source of Greenhouse Gases (GHG). Peatland draining for agricultural purposes is a large contributor to carbon dioxide emissions (e.g., Günther et al., 2020), and permafrost thaw in peatlands is projected to lead to large methane emissions (e.g., Hugelius et al, 2020). Peatlands are thus key environments for climate change mitigation strategies, and there is an urgent need to implement global peatland restoration and protection policies (Beaulne et al., 2021; Günther et al., 2020; Humpenöder et al., 2020). High-quality data on the extent and depth of peatlands is crucial to support decision-making and reach the international goals for sustainable development.

Traditionally, landscape scale peat mapping has been conducted through aerial image interpretation combined with logistically challenging field inventories. Today, automated algorithms for digital mapping with remotely sensed data have made it possible to circumvent these limitations (Minasny et al., 2019). Optical and Synthetic Aperture Radar (SAR) satellite imagery and LiDAR-based terrain indices are used for peatland delineation and depth estimation, often together with auxiliary data on

climate and soil. Limitations for these modern techniques are instead set by data availability, especially LiDAR-based elevation

data and field data for model development.

Sweden presents a good case-study for digital mapping of peatlands as LiDAR-based elevation data is readily available and field data on soil properties has been collected by longstanding nation-wide surveys. In addition, the large extent of Swedish peatlands and their considerable share of drainage, especially in productive forest areas, lead to important GHG emissions warranting further research and mitigation policies (Lindgren & Lundblad, 2014; Skogsstyrelsen, 2021).

Swedish peatlands have been mapped with both traditional and modern methods, with estimates ranging between 15 % and 20 % of the total land area (e.g., Barthelmes et al., 2015; Joosten, 2010; Pahkakangas et al., 2016; Tanneberger et al., 2017; Ågren et al., 2022). Ågren et al. (2022) produced a series of digital Swedish peatland maps at 2 m spatial resolution representing different peat depths. These maps are reported to perform better than traditionally produced maps. Generated by fitting a cubic relationship between O-horizon thickness and a soil moisture product, these maps show greater total peatland cover than

traditional maps due to the inclusion of previously unmapped smaller peatlands. In this study we used partly overlapping O-horizon data, but employed multiple independent variables and machine learning modelling methods to map Swedish forest peatlands.

### 1.1 Study aim

We aimed at mapping the extent of forest peatlands in Sweden, by analysing O-horizon thickness measurements against

LiDAR-based terrain indices. Machine learning methods were applied to construct models able to binary classify shallow peatlands with >= 30, 40 or 50 cm O-horizon thickness and deeper peatlands with >=100 cm O-horizon thickness. Maps of the classifications were produced as 10 m raster grids at a national scale.

## 2 Materials and Methods

### 2.1 Study area

This study was limited to the forested landscapes within the country of Sweden, located on the Scandinavian peninsula. Sweden has a land area of 407,284 km$^2$, of which approximately 69% is covered by forest (Statistiska Centralbyrån, 2022). The country has a pronounced South-North gradient between 55° 20' N and 69° 3' N, with the consequent effect on the climate ranging between temperate and subarctic. The vicinity to the gulf stream and continental west coast setting of Sweden enforce a warmer climate than other areas at similar latitudes. The annual rainfall generally varies between 500 and 800 mm with local extremes

of 400 and 2000 mm (Sveriges Meteorologiska och Hydrologiska Institut [SMHI], 2009). The topography in Sweden is mainly influenced by the Caledonian orogeny, pre-glacial deep-weathering and quaternary glaciations. The bedrock is generally igneous and metamorphic except in the southernmost parts of Scania County, Öland, Gotland and various isolated pockets in mainland Sweden. Soils are predominately of quaternary origins and mostly till, with coastal areas having been influenced by wave action due to isostatic depression (Fredén et al., 2009).

Earth System Science Data Discussions — Open Access

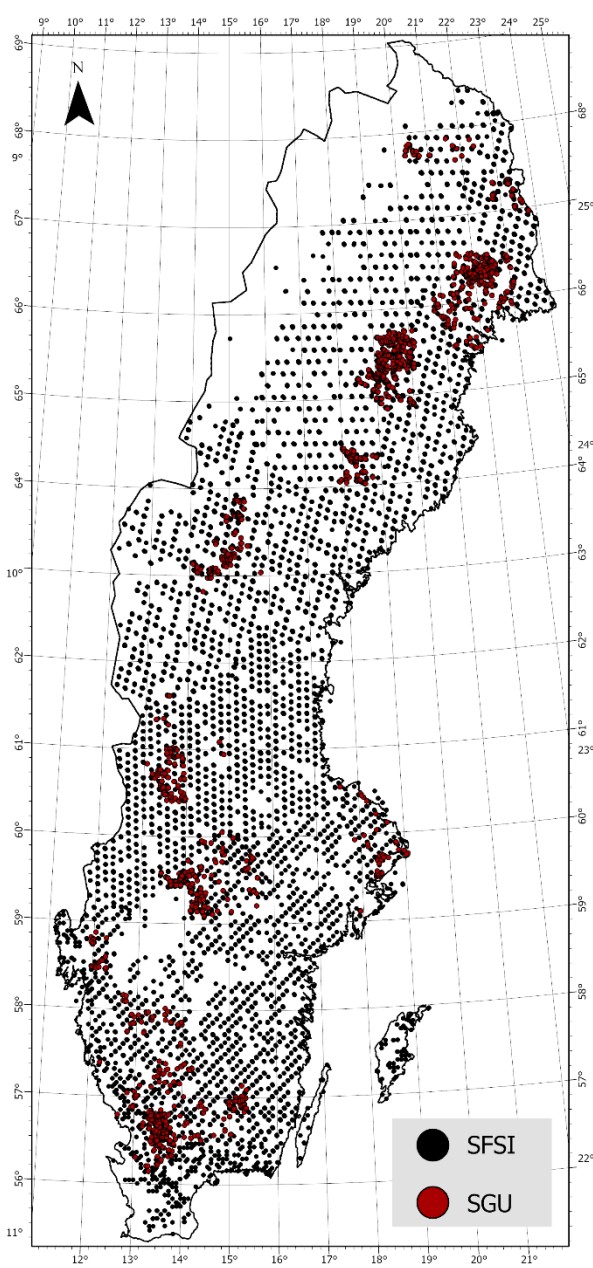

**Figure 1: Location of the O-horizon thickness sampling points included in this study. Black points are part of the Swedish Forest Soil Inventory (SFSI), whereas the red points are from the Peat Core archive by the Geological Survey of Sweden (SGU).**

In this study we only produced maps of peatlands in Swedish forests and treeless wetlands, as the employed O-horizon thickness data was not available in non-forested areas (see sect. 2.2.1). A forest mask was used to delineate areas of interest, consisting of a reclassified version of the Swedish Land Cover dataset from 2018, Nationella Marktäckedata 2018 (NMD2018, Naturvårdsverket). See sect. 2.2.3 for further descriptions.



### 2.2 Data

#### 2.2.1 Dependent variable: O-horizon thickness

A Swedish dataset on O-horizon thickness was constructed by merging two data sources. The first source, the Swedish Forest Soil Inventory (SFSI), is a nationally covering soil survey for which O-horizon thickness has been measured in the range 0-99 cm (Olsson, 1999). The O-horizon is defined as a layer of organic material with >= 20% organic carbon. The SFSI inventories are designed to cover forested land across Sweden, and excludes agricultural areas, urban land, infrastructure and mountain regions.

The second data source was the peat archive of the Geological Survey of Sweden (SGU). Peat core measurements collected by peat experts over the past 100 years are stored in this archive, with varying methods and data quality. Unlike the SFSI no lower limit in peat depth is set.

The accuracy of the geographical positioning is 5-10 m in the SFSI (Olsson, 1999), while for the peat core data from SGU it depends on the measurement period and method. In total, the merged dataset includes 10115 data points, of which 7175 are from the SFSI and 2940 from the SGU peat archive; their spatial distribution can be viewed in fig. 1.

Four thresholds were used to divide the O-horizon thickness dataset into separate binary variables. The first three thresholds were set to 30, 40 and 50 cm. This is the minimum O-horizon thickness of peatlands according to a number of national and international soil classification systems (Lourenco et al., 2022). The last threshold was set to 100 cm, based on the measurement range limitations of SFSI (no data reported >=100 cm). These binary variables (peat deeper or shallower than 30, 40, 50 or 100 cm, respectively) were used separately as dependent variables in machine learning workflows, with the final goal to produce four different geospatial products: maps over peatlands with O-horizon thickness >= 30, 40 and 50 cm and a map over deeper peatlands with O-horizon thickness >= 100 cm.

#### 2.2.2 Features

The independent variables tested for prediction of peatland occurrence, hereby referred to as features, can be seen in table 1. We hypothesised that these features indicate peatland occurrence at landscape scale in Sweden and made the assumption that only LiDAR-based variables were sufficient for prediction. Nonetheless, peat formation and thickness are complexly controlled by climate, hydrology, topography, soil and vegetation (Limpens et al., 2008). In earlier versions of the models presented in this article some variables on climate and vegetation were thus included, but these did not enhance classification performance and were removed.

DEM10 is a LiDAR-based Digital Elevation Model (DEM) with a spatial resolution of 10 m in the geographical coordinate system SWEREF99 (EPSG: 3006). DEM10 is the base-layer of all the other features, except MarkFuktIDX.

*Topographic Position Index* (*TPI*; Weiss, 2001), *Topographic Ruggedness Index* (*TRI*; Riley et al., 1999) and *Roughness* (Dartnell, 2000) describe the elevation heterogeneity of a landscape. *TPI* is the elevation difference between a DEM pixel and the mean of a determined pixel neighbourhood surrounding it; *TRI* is defined as the square root of the sum of the squared





difference between a DEM pixel and each pixel in its 8-cell neighbourhood; *Roughness* is the largest elevation difference between a DEM pixel and a determined pixel neighbourhood.

Slope describes the inclination in a DEM neighbourhood, in our case computed using an algorithm by Horn (1981).

DTW is a topographic index developed by Murphy et al. (2007), approximating the elevation difference between a pixel in a

DEM and a hydrological source. The hydrological sources are defined as pixels with an upslope area larger or equal to a set threshold, the so-called flow initiation area, generally ranging between 0.5-40 ha. Source pixels are allocated for each pixel in a slope raster, applying a least-cost-path algorithm; the accumulated slope values along the path are multiplied by the resolution of the raster to obtain the DTW index. Mathematically, DTW is defined as:

$$DTW = \left[ \sum \frac{dz_i}{dx_i} a \right] x_c$$

where $dz_i$ and $dx_i$ are the vertical and horizontal distance between two cells; $a$ is 1 or $\sqrt{2}$ depending on whether two adjacent cells connect parallelly or diagonally; $x_c$ is the raster cell size in meters.

DTW has been used to identify wet areas with the advantage of being independent of spatial scale (Ågren et al., 2014). The optimal threshold for flow initiation area is however dependent on soil characteristics, as areas with higher hydraulic conductivity are better examined with a higher flow initiation area, and vice versa.

MarkFuktIDX is a Swedish soil moisture index product created support delineation of wetlands in the national land cover dataset (NMD 2018). It was created by merging DTW rasters with Topographic Wetness Index rasters (TWI), with an

**Table 1: Raster datasets tested as features in this study. Features denoted with and asterisk were included in the final models used for peatland mapping, the others were excluded during pre-processing or feature selection (see sect. 2.3.2).**

| *Feature* | *Designation* |
|---|---|
| **Digital Elevation Model** | *DEM10 \** |
| **Topographic Position Index** | *TPI \** |
| **Topographic Ruggedness Index** | *TRI* |
| **Roughness** | *Rough \** |
| **Slope** | *Slope* |
| **Standard Deviation Slope** | *SDSlope \** |
| **Depth to water, 1ha** | *DTW1ha \** |
| **Depth to water, 5ha** | *DTW5ha* |
| **Depth to water, 10ha** | *DTW10ha \** |
| **Depth to water, 20ha** | *DTW20ha* |
| **NMD Markfuktighetsindex** | *MarkFuktIDX \** |





unbalanced weight towards DTW. The DTW raster was calculated using a 2-ha flow initiation area threshold, and the TWI raster included information on soil permeability where this was available.

### 2.2.3 Forest and treeless wetland mask

The forest and treeless wetland mask used to limit the analyses extent in this study consisted of a binary reclassified version of NMD 2018. The positive class included forest classes (codes 111-128) and open wetland (code 2); the negative class (where no analyses were made) includes arable land (code 3), open land (codes 41 & 42) and artificial surfaces (codes 51-53).

### 2.3 Data processing

### 2.3.1 Raster processing

The topographic variables were calculated using a 10 m resolution DEM (DEM10, see table 1), originally resampled with the bilinear method from the 2 m resolution Höjddata 2+ grid DEM from Lantmäteriet.

TPI, TRI, Rough and Slope were calculated using the gdaldem utility (GDAL/OGR contributors, 2022). TPI, TRI, Rough and standard deviation of slope (SDSlope) were calculated on an 8-cell neighbourhood basis.

DTW was calculated with a python script using tools from the Python packages RichDEM (Barnes, 2016) and Whitebox

(Lindsay, 2014). Four different flow initiation area thresholds were used: 1 ha, 5 ha, 10 ha and 20 ha. These were subjectively chosen, but fall within the range of thresholds used in previous studies at similar scale (e.g., Oltean et al., 2016; Ågren et al., 2014; Ågren et al., 2021).

### 2.3.2 Machine Learning workflow

The rasters which constituted the features (see tab. 1) were sampled at the geographical position of each O-horizon data point.

Totally, 10115 data points were extracted. Of these, 4230 were labelled as "peatland" using the 40 cm threshold and 3686 as "deeper peatland" (>=100 cm).

The labelled dataset was pre-processed by imputation, feature correlation analysis, undersampling, dataset splitting and standardization. Most utilities are from the Scikit-learn python package (Pedregosa et al., 2012), hereafter referred to as sklearn. Sklearn SimpleImputer was employed for imputation, replacing missing values in the features with their respective mean. This

process was applied to avoid data loss and because some machine learning algorithms cannot handle missing values.

Features with a Pearson correlation coefficient >= 0.8 between them were then removed from the labelled dataset, as highly correlated features can negatively affect the performance of machine learning algorithms and redundant data enhance processing times and computational costs.

RandomUnderSampler from the imbalanced-learn python package (Lemaitre et al., 2017) was applied to avoid detrimental

class imbalance caused by over-emphasization of the most populous class. This algorithm removes randomly selected samples from the over-represented class to reach complete class balance.

Sklearn train_test_split algorithm was used to randomly split the labelled dataset into a training and a testing dataset, with a 70/30 split ratio. A standardization algorithm, sklearn StandardScaler, was lastly fitted to the features of the training dataset and used to transform the features of all datasets by mean removal and data scaling.

In machine learning, feature selection is performed to find balance between computational cost, model interpretability and model performance (Brank et al., 2011). In this study, the feature selection algorithm RFECV from sklearn was run following pre-processing and prior to model training. RFECV is a wrapper method testing recursively smaller subsets of features by cross-validation and eliminating the weakest feature at each iteration.

Four Machine learning algorithms from the sklearn library were tested, Support Vector Classifier (SVC), Logistic Regression
(LR), Random Forest Classifier (RF) and Multi-Layer Perceptron Classifier (MLP). Sklearn RandomizedSearchCV was used for hyperparameter tuning, which randomly tests subsets of parameters by cross-validation and selects the best performing configuration.

Model configurations were evaluated using accuracy, precision, recall and Matthew's correlation coefficient (MCC) as performance metrics, all computed on the test dataset. These performance metrics were used to identify the most suited model
configurations for final prediction mapping, referred to as Peat30 (O-layer thickness >= 30 cm), Peat40 (O-layer thickness >= 40 cm), Peat50 (O-layer thickness >= 50 cm) and Peat100 (O-layer thickness >= 100 cm). We regarded high precision as a positive model trait, as for peatland management we believe it is relatively more important to minimize false positives rather than false negatives. Therefore, if two models had similar accuracy and MCC, we prioritized the one with higher precision.

Permutation feature importance was calculated for each final model, a measure of the impact of each feature on predictive
power. Permutation feature importance is calculated as performance lost, in our case accuracy loss, as the feature is permuted (i.e., shuffled) in the test dataset. Hereafter it is referred to as feature importance.

Final prediction mapping

Final predictions were executed with Peat30, Peat40, Peat50 and Peat100 on a raster grid of 10 m resolution, corresponding to the raster variables used as features (see tab. 1). Four maps where thus produced, three for peatlands (O-layer thickness >= 30,
40 or 50 cm) and one for deeper peatlands (O-layer thickness >= 100 cm). Some areas in the mountainous regions are missing in the final products, as LiDAR data is not available here.

Non-forested land was masked in the final maps using the mask described in sect. 2.2.3. The maps were compressed using the gdal_translate Lempel-Ziv-Welch compression algorithm and overview images were built internally using gdaladdo (GDAL/OGR contributors, 2022).
Map visualizations were created using ESRI ArcGIS Pro 2.9.2.

## 3 Results

Prior to model training, the same features were selected by the selection algorithm RFECV in all classification tasks: DEM10, DTW1ha, DTW10ha, Rough, SDSlope, TPI, MarkFuktIDX.





The test results are shown in tab. 2, demonstrating that RF and MLP have marginally higher accuracy and MCC than LR and

SVC in most cases.

In the 30 cm classification task the RF model scored the highest MCC and accuracy values with balanced precision and recall values, thus being selected as Peat30 (tab. 2a). In the 40 and 100 cm classification tasks the RF and MLP models scored the same performance metrics (tab. 2b, 2d); however, for the sake of consistency between the final maps, the RF models were selected as Peat40 and Peat100. In the 50 cm classification task the SVC and RF models scored the same accuracy and MCC

values; however, the RF model scored higher precision, thus being selected as Peat50 (tab. 2c).

**Table 2: Performance metrics computed on the test set for the models operating on the various binary O-layer thickness thresholds: (a) 30 cm, (b) 40 cm, (c) 50 cm, (d) 100 cm. Asterisk and bold numbers denote the models selected for final prediction mapping, Peat30, peat40, Peat50 and Peat100. LR=Logistic regression, SVC=Support Vector Classifier, RF=Random Forest MLP=Multi-layer Perceptron**

| (a) | LR | SVC | RF* | MLP |
|---|---|---|---|---|
| *Accuracy* | 0.89 | 0.89 | **0.90** | 0.89 |
| *Precision* | 0.86 | 0.87 | **0.90** | 0.89 |
| *Recall* | 0.93 | 0.91 | **0.90** | 0.89 |
| *MCC* | 0.78 | 0.78 | **0.79** | 0.78 |

| (b) | LR | SVC | RF* | MLP |
|---|---|---|---|---|
| *Accuracy* | 0.90 | 0.90 | **0.91** | 0.91 |
| *Precision* | 0.88 | 0.89 | **0.91** | 0.91 |
| *Recall* | 0.93 | 0.92 | **0.91** | 0.91 |
| *MCC* | 0.80 | 0.81 | **0.81** | 0.81 |

| (c) | LR | SVC | RF* | MLP |
|---|---|---|---|---|
| *Accuracy* | 0.89 | 0.90 | **0.90** | 0.89 |
| *Precision* | 0.86 | 0.88 | **0.90** | 0.89 |
| *Recall* | 0.92 | 0.93 | **0.91** | 0.90 |
| *MCC* | 0.78 | 0.80 | **0.80** | 0.79 |

| (d) | LR | SVC | RF* | MLP |
|---|---|---|---|---|
| *Accuracy* | 0.88 | 0.89 | **0.89** | 0.89 |
| *Precision* | 0.83 | 0.85 | **0.87** | 0.87 |
| *Recall* | 0.94 | 0.94 | **0.92** | 0.92 |
| *MCC* | 0.76 | 0.78 | **0.79** | 0.79 |

Permutation feature importance for all final models can be viewed in fig. 2. The two most prominent features were the same in all models, MarFuktIDX and Rough. In Peat100 MarFuktIDX and Rough showed highly similar results, compared to the other models showing larger gaps between them (fig. 2). It is worth noting that a low feature importance does not necessarily mean that the feature is insignificant for a given classification task. Features with low feature importance in one model can be

of greater importance in a differently configured model trained on the same data.

Excerpts of the final prediction map for Peat40 can be viewed in fig. 3 and 4. Peat areas clearly follow topographic basins and shallow peat often surrounds areas of deeper peat or is found in smaller patches and furrows. The estimated national extents of peatlands according to Peat30, Peat40, Peat50 and Peat100 can be viewed in table 3.

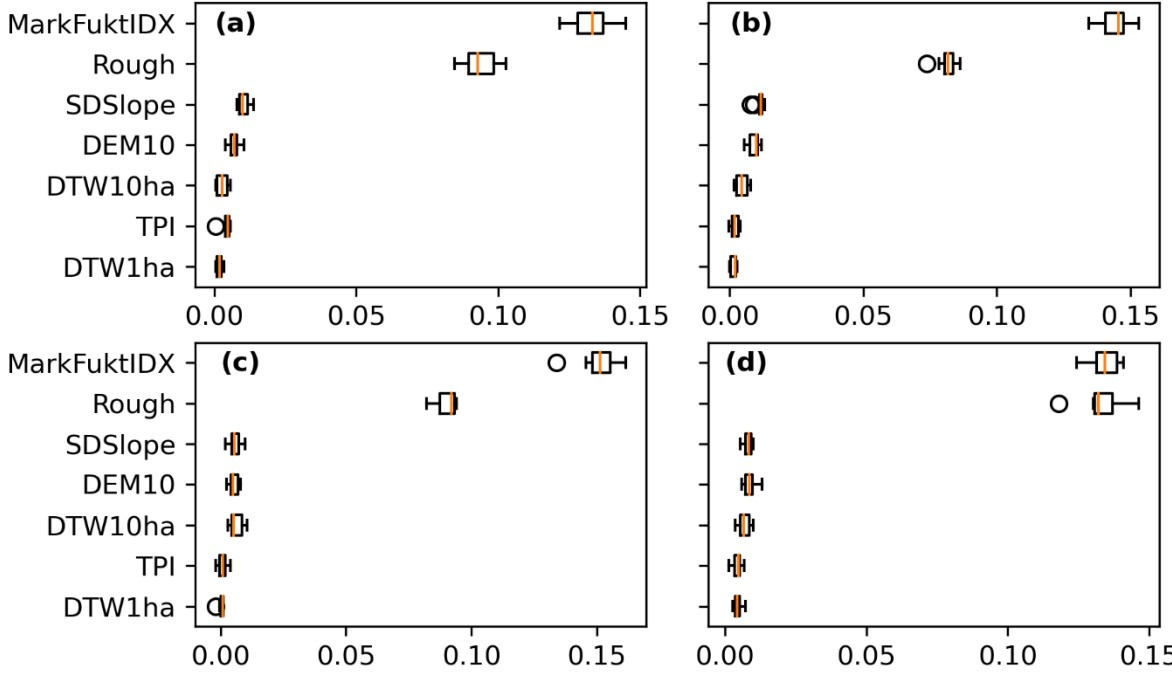

**Figure 2: Boxplots showing the permutation feature importance of (a) Peat30, (b) Peat40, (c) Peat50 and (d) Peat100. Permutation feature importance is calculated as the loss of accuracy when a feature is permuted, in this case iterated ten times. The boxes represent the lower and upper quartiles with an orange line at the median; whiskers indicate data ranges and isolated point are outliers.**

**Table 3: Estimated national extents of peatlands in forested and treeless wetland areas according to the models developed in this study. %$_{Tot}$ is the percentage of extent referenced to the total Swedish land area (407,284; Statistiska Centralbyrån, 2022); %$_{Forest}$ is the percentage of extent referenced to the Swedish forested and treeless wetlands area (NMD 2018 Forest and treeless wetland mask, see section 2.2.3.).**

| Extent | Peat30 | Peat40 | Peat50 | Peat100 |
|---|---|---|---|---|
| $km_2$ | 72,604 | 67,260 | 66,378 | 60,726 |
| %$_{Tot}$ | 17.8 | 16.5 | 16.3 | 14.9 |
| %$_{Forest}$ | 23.5 | 21.8 | 21.5 | 19.7 |







**Figure 3: Excerpt from the final *Peat40* map. Orange areas are classified as having an O-horizon >= 40cm. The map covers an area with extensive Aapa mire complexes in Udtja, Norrbotten county, Sweden. Base map: Ortofoto, 0.5 RGB Tiles 735_67_50_2014 and 736_67_00_2014 © Lantmäteriet (2014).**




**Figure 4: Excerpt from the final *Peat40* map. Orange areas are classified as having an O-horizon >= 40cm. In the center of the map is an area with an extensive Fen complex in Kråketorp, Kronoberg county, Sweden. Base map: Ortofoto, 0.5 RGB Tile 634_48_05_2011 © Lantmäteriet (2011).**



## 5 Discussions

This study is product-driven, with the aim to map the extent of peatlands in the forested landscape of Sweden. The performance metrics (tab. 3) indicate that the models we developed have good predictive power and that the resulting maps are of adequate

quality. However, the results for Gotland and Öland may not to be entirely reliable: these islands were scarcely sampled with 64 sampling points in Gotland and 16 points in Öland, of which only 1 point per island was classified as peatland using O-layer thickness >=40 cm as definition. The signal from the sampling on the mainland could therefore have had adverse influence on the quality of prediction on Gotland and Öland, which have relatively distinct geological characteristics in relation to mainland Sweden.

It would have been beneficial to have extensive O-layer thickness measurements from all land cover types, in which a land cover-based mask would not have been needed and the resulting maps could include all land cover types. We also note that a mask of forest areas based on SFSI would have been more consistent with the utilized soil data than a reclassified version of NMD 2018, but it was unavailable at the time of this study.

We hypothesize that the greater importance attributed to *Rough* in *Peat100* (fig.2) is due to deeper peat being generally more

concentrated to the inner and flat parts of peatlands. The over-arching importance of *MarfuktIDX* (fig.2) we believe is caused by it being a high-quality descriptor of soil moisture, an essential edaphic factor for peat formation.

Our estimates of national forest peatland extent are within the range of studies employing traditional peatland maps for data merging and upscaling (Barthelmes et al., 2015; Pahkakangas et al., 2016; Tanneberger et al., 2017). These studies were based on maps of quaternary deposits by SGU and estimated that 62,073-69,155 km2 of Sweden is covered by peat; the range is

caused by the utilization of different techniques to fill the gaps in the SGU maps and by using or not the SGU map with coarsest resolution. These estimates are however not entirely comparable to ours, as SGU does not exclude non-forested land as us. Also, the SGU maps often exclude smaller peatlands by instead reporting the dominant soil type in a given area (Karlsson et al., 2021), probably leading to an overall underestimation of peatland extent.

Assuming that agricultural land on peat is around 2,300-2,500 km2 (Berglund & Berglund, 2010; Lindahl & Lundblad 2021)

and that the area of wetlands in the mountainous areas with no LiDAR scans is approximately 650 km2 (Wetland class in NMD2018, Naturvårdsverket), our national peatland extent estimates are similar to the higher end in the range of the traditional maps (e.g., Barthelmes et al., 2015).

On the other hand, Ågren et al. (2022) produced 2 m resolution peatland maps using modern automated mapping methods. They estimated that 70,000-94,000 km2 of Sweden is covered by peat, depending on peatland definition, of which 68,000-

88,000 km2 is in forests. If defining peat as having an O-layer thickness >= 40 cm, the study estimated 79,000 km2 total peatland extent and 76,000 km2 forest peatland extent, considerably higher estimates than ours. The authors concluded that the difference in estimates between their study and previous ones were caused by the inclusion of smaller peatlands associated to streams and narrow pits not included in the SGU maps. The difference between our estimates of total peatland extent and those of Ågren et al. (2022) is mainly because they do not exclude agricultural land from their final products, as we did.

On our test dataset, the 40 cm peatland map by Ågren et al. (2022) achieved Accuracy, Precision, Recall and MCC values of 0.93, 0.93, 0.93 and 0.86. Comparatively, Ågren et al. (2022) employed a simpler model algorithm than us but obtained better performance metrics. We believe that these results can be deduced to a high-quality soil moisture map being the foundation of their model, supported by our results indicating that soil moisture datasets are the most important DEM-derived products for peatland delineation. Furthermore, the considerably higher performance metrics reported in this paper compared to Ågren et

al. (2022) are probably due to the inclusion of the SGU peat core archive as input dataset, which is skewed towards easily classifiable areas of deep peatland.

## 5 Conclusions

Peatland area, as mapped in this study, is a crucial variable for estimating C stock (Yu, 2012). In turn, such estimates are

important for understanding global C cycles and calculating GHG emissions from peatlands (Beaulne et al., 2021; Hugelius et al., 2020). Our peatland maps may therefore be used for estimating the C stock of Swedish peatlands, for example by coupling them with information on peat thickness and C density. This is especially important in the context of land cover and climate change, which may further contribute to peatland degradation and oxidation (Hugelius et al., 2020; Humpenöder et al., 2020). In conclusion, our results highlight the possibilities of producing boreal peatland maps using LiDAR-based terrain indices and

spatially and numerically extensive field data. We have demonstrated that an approach such as ours, readily replicable and using only open-source software, can yield accurate and uniform products surmounting the limitations of traditional mapping techniques. Ågren et al. (2022) and Karlson et al. (2019) also concluded that terrain indices are highly valuable for peatland mapping in Sweden; we therefore emphasize that the availability of high-quality LiDAR and field data is a cornerstone to improve mapping of boreal peatlands. Such advancements could in turn reduce the uncertainties in C stock estimations

attributed to poor data on peatland extent, and thus improve estimates of GHG fluxes. Information on the location and depth of peatlands is also crucial for protection and restoration initiatives, making high quality geospatial products essential for climate change mitigation strategies and decision-making.

## 6 Data availability

The peatland maps Peat30, Peat40, Peat50, Peat100 are available as GeoTIFF files in the Bolin Centre Database

(https://doi.org/10.17043/rimondini-2023-peatlands-1, Rimondini et al., 2023).



**Author contributions**

Lukas Rimondini and Gustaf Hugelius designed the study. Anders Ahlström contributed with the SFSI data. Lukas Rimondini performed the data collection, analysis and curation. The manuscript was written by Lukas Rimondini with inputs from Gustaf Hugelius and Thomas Gumbricht.

**Competing interests**

The authors declare that they have no conflict of interest.

**Acknowledgements**

We thank the Swedish Transport Administration, Trafikverket, for funding part of this work through their FOI project "Våtmarkers ekosystemtjänster: påverkan från nya projekt och möjligheter till klimatkompensation" (Project number: TRV 265 2020/118654). Furthermore, we thank contributing staff at SLU and SGU for making the soil data employed in this study available.

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
