# Peer review of "Mapping of peatlands in the forested landscape of Sweden using LiDAR-based terrain indices"

_Earth System Science Data, 2023_

## Referee Comment (RC1)

essd-2023-77    Submitted on 03 Mar 2023
Mapping of peatlands in the forested landscape of Sweden using LiDAR-based terrain indices
Lukas Rimondini, Thomas Gumbricht, Anders Ahlström, and Gustaf Hugelius

The manuscript presents the use of machine learning to model soil data against LiDAR-based terrain indices for the purpose of producing national scale peat depth maps for Sweden. The authors use data acquired from the Swedish Forest Soil Inventory and the Geological Survey of Sweden alongside LiDAR derived DEM. The combined soil data provide good coverage and data gaps are explained. Assumptions and decisions made on the data used are also well reasoned. The independent variables, referred to as features are topographic classifications derived from the DEM. A reference for each is provided so the reader could find further detail.

The work is valuable in that it presents a technique that is relatively simple to reproduce across other nations (where equivalent soil and LiDAR datasets are available). In all sections the information presented is clear and concise. The data sources are well explained and the data processing steps are outlined in an instructive and accessible way so that the reader could repeat on their own datasets.

The model performance and results are well reasoned, and the example images are nicely presented. However, I found the comments in the Discussion regarding the reliability of Gotland and Oland results a little confusing. Would it not have been better to exclude these areas?

The explanation for data download was simple and clear. Data was downloaded (from https://bolin.su.se/data/rimondini-2023-peatlands-1) and each of the 4 raster datasets was viewed in QGIS with ease and appeared to be complete and of high quality.

My overall recommendation would be to accept the manuscript with the following minor corrections addressed.

| Abstract, Line 10 | *degradation by land cover* should read *degradation by land cover* **change** |
|---|---|
| 2.2.2 Features, Line 115 | *created support delineation* should read *created* **to** *support delineation* |
| Table 1 caption | *with and asterisk* should read *with* **an** *asterisk* |
| 2.3.2 Machine Learning workflow, Line 135 | *Totally, 10115 data points* would read better as **In total**, *10115 data points* |
| Line 167 | *Final prediction mapping* Unclear if this is supposed to be a sub-heading? |
| Table 2 caption | Capitalise *peat40* to make consistent with other mapping layer titles. |
| 5 Discussions, Line 205 | *in which a land* should read *in which* **case** *a land* |
| Line 214 | *Superscript missing Km2* |
| Line 216 | *land as us* would read better as *land as* **we did** |
| Lines 219, 220, 224, 225, 225 | *Superscript for km2* |

---

## Author Response (AR1)

The line numbers reference the marked-up document, Rimondini_etal_2023_Manuscript_Ver4_TrackChanges.pdf

**Referee 1**

- **Comment 1**: *"However, I found the comments in the Discussion regarding the reliability of Gotland and Oland results a little confusing. Would it not have been better to exclude these areas?"*

  **Response 1:** We agree that the product should exclude Gotland and Öland. We propose that they should be masked in the next version and that the paragraph regarding the reliability of their mapping should be lightly revisioned and moved from "Discussions" to "Methods". National peat estimates are changed accordingly.

  **MS changes 1:**

  - Line 16: Change national peat estimates
  - Line 172-175**:** Add *"Gotland and Öland were also excluded in the final maps. These islands were scarcely sampled with 64 sampling points in Gotland and 16 points in Öland, of which only 1 point per island was classified as peatland using O-layer thickness >=40 cm as definition. The signal from the sampling on mainland Sweden could therefore have adverse influence on the quality of prediction on these islands, which have relatively distinct geological characteristics."*
  - Line 230-231: Add *"and on Gotland and Öland 150 $Km_2$ together"*
  - Table 3: Change national peat estimates, subtracting Öland and Gotland
  - Line 266 & 359: Change reference to dataset version 2

- **Comment 2:** *"My overall recommendation would be to accept the manuscript with the following minor corrections addressed."*

  **Response 2:** We appreciate the corrections you pointed out and will change the content of the manuscript accordingly.

  **MS changes 2:**

  - Line 10: add *"change"*
  - Line 115: add *"to"*
  - Table 1 caption: change *"and"* to *"an"*
  - Line 135: change *"Totally"* to *"In total"*
  - Line 167: adapt to sub-heading and add *"2.3.3"*
  - Table 2 caption: capitalize *"peat40"*
  - Line 215: add *"case"*
  - Lines 224, 229, 230, 234, 235, 236: change *"km2"* to *"$Km_2$"*
  - Line 227: change *"us"* to *"we did"*

**Referee 2**

- **Comment 3**: *"It is mentioned that the reference data has some spatial inaccuray. Unfortunately, this topic is no longer covered in the discussion. It is strongly recommended to discuss the potential influence of the spatial inaccuracies on the derived result."*

  **Response 3**: We agree that it would be of interested to further discuss how the spatial inaccuracy may affect the results. In brief, we believe that the spatial inaccuracy of the soil data mostly concerns the SGU peat archive. We believe that it is of minor importance for our results, as the data from the SGU archive mostly includes points from the inner parts of peatlands with an O-layer depth > 100 cm and a relatively narrow value range in the geodata used as features. We have visually checked if this is the case by comparison against a land cover dataset (NMD2018), and conclude that ~100 points part from this pattern (mostly in the Stockholm area). This means that only a small portion of the points may lead to any important errors in the features. This will be reflected in the updated manuscript.

  **MS change 3** (Line 204-214)**:** Remove previous text about Gotland and Öland and add *"Some errors in the features may have been caused by the spatial inaccuracies of the soil data, especially in the case of the SGU peat archive which is a collection of historical data with varying quality. However, we believe that this has had minor effect on our results, as the data from the SGU archive mostly includes points from the inner parts of peatlands with an O-layer depth > 100 cm and a relatively narrow value range in the geodata used as features. Only a minority of the SGU archive data points around Stockholm part from this pattern, meaning that a small portion of the points may lead to any important errors in the features."*

- **Comment 4: "**For the used formulas, please add unit information to all used variables."*

  **Response 4:** We appreciate your comment on the inclusion of unit information for all variables in the formulas. We interpret that your comment concerns the DTW formula, and will add the information accordingly in the next version of the manuscript.

  **MS change 4** (Line 110-111): "*where $dz_i$ and $dx_i$ are the vertical and horizontal distance in meters between two cells; a is a constant of 1 or √2 depending on whether two adjacent cells connect parallelly or diagonally; $x_c$ is the raster cell size in meters.*"